# PROCAT:
# Product Catalogue Dataset for Implicit Clustering, Permutation Learning and Structure Prediction

**Mateusz Jurewicz**\*
Department of Computer Science
IT University of Copenhagen
København, 2300
`maju@itu.dk`

**Leon Derczynski**
Department of Computer Science
IT University of Copenhagen
København, 2300
`leod@itu.dk`

## Abstract

In this dataset paper we introduce PROCAT, a novel e-commerce dataset containing expertly designed product catalogues consisting of individual product offers grouped into complementary sections. We aim to address the scarcity of existing datasets in the area of set-to-sequence machine learning tasks, which involve complex structure prediction. The task's difficulty is further compounded by the need to place into sequences rare and previously-unseen instances, as well as by variable sequence lengths and substructures, in the form of diversely composed catalogues. PROCAT provides catalogue data consisting of over 1.5 million set items across a 4-year period, in both raw text form and with pre-processed features containing information about relative visual placement. In addition to this ready-to-use dataset, we include baseline experimental results on a proposed benchmark task from a number of joint set encoding and permutation learning model architectures.

## 1   Introduction

Intelligent product presentation systems and catalogue structure prediction are important areas of research, with clear practical applications [de Melo et al., 2019] and a substantial impact on the environment [Liu et al., 2020]. With the ultimate goal being the reduction of paper waste stemming from print catalogues, in this paper we present a dataset of over 10,000 catalogues consisting of more than 1.5 million individual product offers. This dataset lends itself to machine learning research in the area of set-to-sequence structure prediction, clustering and permutation learning.

Whilst there are many e-commerce product datasets containing information about individual product offers for the purposes of recommendation [Fu et al., 2020] and categorization [Lin et al., 2019], there is a scarcity of publicly-available, easily accessible and reliably maintained product datasets for catalogue structure prediction and permutation learning. Providing such a dataset can help foster the transition from print to digital catalogues [Wirtz-Brückner and Jakobs, 2018].

This task is challenging for machine learning methods due to the necessity of learning to obtain useful representations of **rare and unseen instances** of product offers, the **variable offer and catalogue lengths**, as well as the **implicit clustering task** necessary for predicting the split of offers into a **varying number of clusters** (sections) to output the final catalogue structure.

With this work, we aim to address this domain lacuna in three ways. First, we provide a large dataset of product catalogues designed by marketing experts. These are structured, and the task over them is to predict a catalogue structure given a set of product offers (the set items). This structure takes the form of grouping product offers into complementary sections and ordering or *permuting* the

---

\*Affiliated with the Tjek A/S Machine Learning Department (København, 1408), contact via `mj@tjek.com`.

35th Conference on Neural Information Processing Systems (NeurIPS 2021) Track on Datasets and Benchmarks.

sections into a compelling catalogue narrative [Szilas et al., 2020], a currently qualitative aspect of human-performed task.

Second, we perform a series of experiments on this dataset, obtain initial benchmarks of performance and propose a number of combined set-to-sequence model architectures. These architectures, along with all model parameters, are also made publicly available, along with a repository containing all code necessary for repeated experiments.

Third, we supplement the real-world catalogue data with a code library for generating simplified, automatically-synthesized product catalogues that adhere to flexible, adjustable structural and distributional rules. These synthetic catalogues can then be used to train set-to-sequence structure prediction models analogous to the ones we benchmark on the main dataset. Additionally, the library allows for detailed functional metrics on the performance of these models, grouped into specific aspects of the chosen structural rules. This allows for greater insight into what kinds of structures different types of models are effective at learning and full control over the task's difficulty.

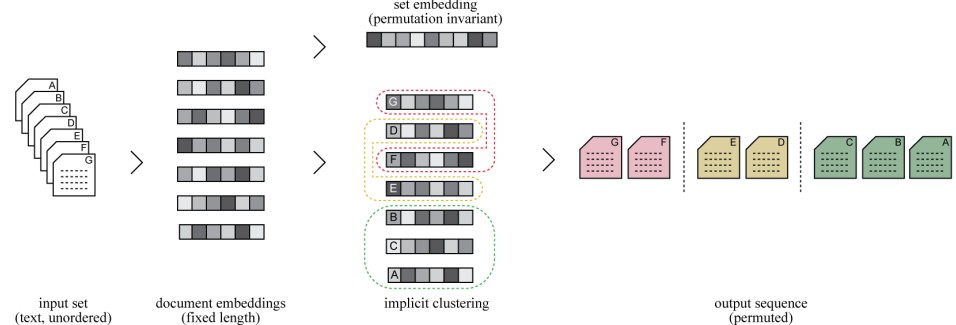

Figure 1: Diagram visualizing the core set-to-sequence structure prediction task through permutation learning with implicit clustering and set representation learning.

The remainder of this paper is structured in the following way: in section 2 we elaborate on prior work, existing datasets and relevant structure prediction methods in more detail. In section 3 we introduce the specifics of the main dataset contribution, including data collection, composition, pre-processing, distribution and ethical considerations. For further details regarding the dataset see the datasheets for datasets checklist [Gebru et al., 2018] in section A.3 of the appendix. In subsection 3.4, we outline the synthetic dataset generation library and its related functional testing capacities. We then move on to section 4, where the experimental setup and initial benchmark results are presented. Finally, sections 5 and 6 discuss the limitations of our work and conclusions respectively, with minor notes on the potential for future work.

## 1.1 Our contributions

- PROCAT dataset of over 10,000 human-designed product catalogues consisting of more than 1.5 million individual product offers, across 15 GPC commercial product categories.
- Library for generating simplified, synthetic catalogues according to chosen structural rules and measuring related model performance through functional tests, with full control over the task's difficulty.
- Benchmark evaluation tasks and baseline results for 4 proposed deep learning models utilizing both datasets.

The links to all mentioned resources including the PROCAT dataset, the code repository for repeated experiments and the best performing model weights are provided in the appendix, in subsection A.1.

## 2 Prior work

Research interest into the process of digitizing paper product catalogues into internet-based electronic product catalogues (IEPCs / EPCs) has a long history [Palmer, 1997, Stanoevska-Slabeva and Schmid, 2000, Guo, 2009, de Melo et al., 2019]. There are ample machine learning datasets consisting

of individual products [Xiao et al., 2017] or product reviews [Haque et al., 2018], but excluding information about the structure of a readable catalogue composed from such offers. To the authors' knowledge, no publicly available dataset containing both the features of individual product offers and the order and grouping in which they were presented as a product catalogue exists.

In order to empower more businesses to present their available products in a visually pleasing digital form and move away from wasteful paper-based solutions, an automatic method for turning a set of offers into a structured presentation needs to be obtained [Guo, 2009]. We propose a set-to-sequence formulation of this task, enabling machine learning models to learn the optimal structure of a viewable product catalogue from historic examples.

With that framing of the task in mind, a very brief overview of existing set-to-sequence, permutation learning model architectures and datasets is given below.

## 2.1 Set-to-sequence methods

Machine learning set-to-sequence methods can approximate solutions to computationally expensive combinatorial problems in many areas. They have been applied to learning competitive solvers for the NP-Hard Travelling Salesman Problem [Vinyals et al., 2015]; tackling prominent NLP challenges such as sentence ordering [Wang and Wan, 2019] and text summarization [Sun et al., 2019]; and in multi-agent reinforcement learning [Sunehag et al., 2018]. A notable example is the agent employed by the AlphaStar model, which defeated a grandmaster level player in the strategy game of Starcraft II, where set-to-sequence methods were used to manage the structured, combinatorial action space [Vinyals et al., 2019]. For a survey of set-to-sequence in machine learning, see Jurewicz and Derczynski [2021].

These model architectures often obtain a meaningful, permutation-invariant representation of the entire available set of entities [Zaheer et al., 2017], either through adjusted recurrent neural networks [Vinyals et al., 2016] or transformer-based methods [Lee et al., 2019]. This is then followed by a permutation learning module whose output is conditioned on the above-mentioned representation. Such modules can take many forms, ranging from listwise ranking [Ai et al., 2018], through permutation matrix prediction [Zhang et al., 2019] to attention-based pointing [Yin et al., 2020].

## 2.2 Set-to-sequence datasets

In lieu of domain-specific datasets for product catalogue structure prediction through set-to-sequence permutation learning, we can look to other areas of machine learning research where predicting a permutation is the goal. These include sentence ordering [Cui et al., 2018], where any source of consecutive natural language sentences can be used, such as the NIPS abstract, AAN abstract, NSF abstract datasets [Logeswaran et al., 2018]. However, this formulation precludes the model from learning an implicit clustering.

Furthermore, sequential natural language tasks such as sentence continuation are fundamentally different from catalogue structure prediction because word tokens come from a predefined vocabulary, whereas new offers may have never been seen before by our models, presenting a further challenge.

Alternatively, one can look to learn-to-rank datasets from the domain of information retrieval, such as Istella LETOR[1] or MSLR30K[2], as used for permutation learning by Pang et al. [2020]. However, learn-to-rank frameworks presuppose an existence of a *query* for which a relevance rating is assigned to each document, which are then sorted according to this rating. It is unclear what could constitute the query in the context of product catalogue structure prediction. The permutation invariant representation of the entire set of available offers is a possible candidate, requiring further research, as mentioned in the conclusion section (6).

Finally, there exist ways to obtain visual permutation datasets consisting of image mosaics, where the task is to reorder the puzzle pieces back into the original image. Santa Cruz et al. [2018] obtain these mosaics from the Public Figures and OSR scene datasets [Parikh et al., 2012]. This resembles the product catalogue prediction task in terms of permuting previously unseen atomic instances (image fragments), but lacks the element of implicit clustering into meaningful, complementary sections.

---

[1]http://blog.istella.it/istella-learning-to-rank-dataset/

[2]http://research.microsoft.com/en-us/projects/mslr/

Table 1: Sample PROCAT offers with raw text features

| section | header | description | priority |
|---------|--------|-------------|----------|
| 1 | Lamb chops | Approx. 400 grams. Marinated chops with mushrooms, bacon. Best served with cream. | A |
| 1 | Ham roast | 700-800 grams. Oriental. Mexico. | B |
| 1 | Melon | Organic piel de sapo or cantaloupe melon. Unit price 20.00. Spain, 1st class. | C |
| 2 | Hair spray | ELNETT. Extra strong. Strong hold. 400 ml. | A |
| 2 | Deodorants | Spray. Roll-on. 50-150 ml. REXONA | B |

## 3 PROCAT

In order to mitigate the lack of product catalogue datasets, with the prediction target being a complex permutation requiring implicit clustering, we propose a new dataset further referred to as PROCAT.

This dataset consists of 11,063 human-designed catalogue structures, made up of 1,613,686 product offers with their text features, grouped into a total of 238,256 sections. The dataset's diversity stems from the catalogues covering 15 different GPC-GS1 commercial categories and from their original composition being created by 2398 different retailers, including cross-border shops that have a significant following in Denmark and neighboring Scandinavian countries, particularly Sweden and Norway, as well as Germany. For more details, see A.2.

What follows is a more in-depth look into the collection and content of this data. For an introductory excerpt demonstrating sample offers from the same catalogue through raw text features, section assignment and priority class, see table 1.

Additionally, we briefly introduce a supplementary library for generating simpler, synthetic structures meant to resemble product catalogues in section 3.4.

### 3.1 Data collection

The data was acquired by Tjek A/S, a Scandinavian company helping people do their shopping, through Tjek's proprietary system for extracting offers from any PDF. This system reads the feeds and scrapes a list of stores and PDF catalogs associated with said stores. Afterwards, a human curation step is performed by the operations department to make sure the obtained data is correct.

The data was collected within the full 4 year period between 2015 and 2019. The original structure of each catalogue is preserved through retaining information about which offers were presented together on which section (page), what the order of sections was and through a separate feature referred to as *priority class*, which represents the relative size of the corresponding offer's image on the page in the original catalogue. A visual representation is given in figure 2.

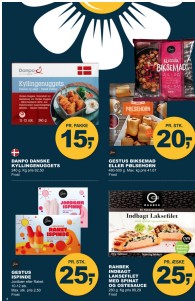 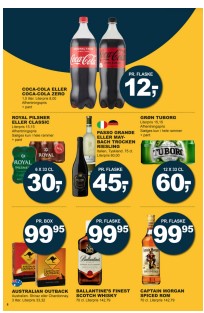 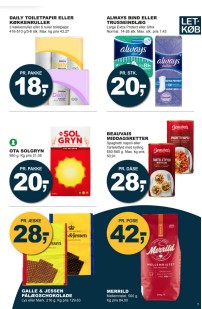

Figure 2: Product offers grouped into 3 consecutive sections extracted from a single catalogue.

### 3.2 Catalogue data

The dataset consists of instances representing 3 types of entities. The most atomic entity is an *offer*, which represents a specific product with a text heading and description, which often includes its on-offer price. Individual product offers are then grouped into *sections*, which represent pages in

a physical catalogue brochure. Finally, an ordered list of sections comprise a single *catalogue*, for which a prediction about its optimal structure is made. This takes the form of permuting the input set of offers into an ordered list, with section breaks marking the start and end of a section.

Each offer instance consists of its unique id, its related section and catalogue ids, a text heading and description in both raw form and as lowercase word tokens obtained via the nltk tokenizer [Bird, 2006], the total token count, and finally the full offer text as a vector referencing a vocabulary of the most common 300 thousand word tokens. Additionally, each offer is categorized into a priority class, representing how visually prominent it was in the original catalogue in terms of relative image size (on a 1-3 integer scale).

Each catalogue instance consists of its unique id, an ordered list of associated section ids, and an ordered list of offer ids that comprise the catalogue in question, including section break markers. Additionally, each catalogue instance also includes information in the form of ordered lists of sections, each containing a list of offers as vectors, with their corresponding priority class and the catalogue's length as the total number of offers within it. Finally, a randomly shuffled *x* of offer vectors (with section breaks) is provided for each catalogue, along with the target *y* representing the permutation required to restore the original order.

Every catalogue instance consists of both raw data and pre-processed features. The dataset is not a sample, it contains all catalogue instances from the years 2015 - 2019 available for viewing in the Tjek A/S app. No other selection filter was used. For a more detailed look at the structure and format of the files comprising the dataset, please see the code repository linked in the appendix in section A.1.

## 3.3   Sustainability

The dataset is made publicly available under the CC BY-NC-SA license. It is hosted by *figshare*, an open access repository where researchers can preserve and share their research outputs, supported by Digital Science & Research Solutions Ltd. The platform was chosen due its prominence, provision of a persistent identifier and rich metadata for discoverability. The dataset will be continuously maintained by the authors of this paper, who can be contacted via the emails provided in the contact information above the abstract.

If labeling errors are found, they will be corrected. The dataset may be expanded with further instances, depending on the academic interest. All previous versions of the dataset will continue to be available. Others are encouraged to extend the dataset and can choose to do so either in cooperation with the authors or individually, in accordance with the chosen license.

## 3.4   Synthetic data and functional testing

In order to experimentally demonstrate the initial viability of model architectures on the type of structure prediction task presented by the product catalogues, we also propose a library for generating simpler, synthetic catalogue datasets. Additionally, we enable researchers to use this library to easily specify hand-picked distributional, structural and clustering rules that determine what kinds of synthetic catalogues are generated. Finally, we provide tooling for obtaining detailed metrics regarding the models' performance per specified rule.

The synthetic datasets also allow for predicting multiple valid catalogue structures from the same underlying input set, which addresses an important limitation of the main dataset, where only one target permutation is available.

The main difference between the real and synthetic datasets is that the basic building block of a catalogue in the latter case takes the form of a vocabulary-based token representing a single product offer. This circumvents some of the difficulty related to representation learning in a few and zero shot setting inherent to the main PROCAT dataset. It becomes natural to think of each offer as representing a member of a wider, colour-coded class, such as green for vegetables, red for meats and so forth. For a visual example see figure 3.

The chosen clustering and structural rules can include pairwise and higher-order interactions between offer types. For example, the presence of both a green and purple offer type in the initial available set can result in a rule which forces the catalogue to be opened with an all-purple section and closed with

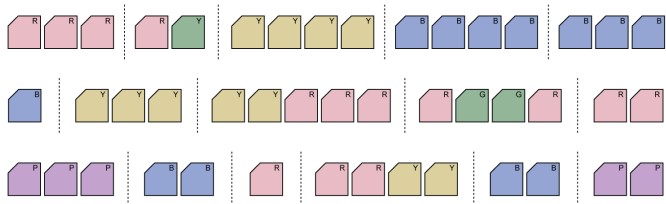

Figure 3: Three synthetic catalogue sequences, consisting of instances of 5 colour-coded offer types, separated into sections and ordered according to chosen distributional, clustering and structural rules. a mixed red and yellow section. The presence of all three primary colours can make a mixed purple and blue section invalid, forcing these offers to be split between two separate sections and so forth.

The ability to obtain structure prediction accuracy metrics per rule enables us to, for example, experimentally test the ability of models such as the Set Transformer [Lee et al., 2019] to encode such higher order interactions in various controlled settings.

# 4 Benchmark task and results

The data provided in PROCAT can motivate a number of benchmarking tasks related to representation learning, clustering, catalogue completion and structure prediction. We focus on a permutation learning approach to predicting the proper structure of a product catalogue, with implicit clustering of the provided set of offers into varying-length sections.

## 4.1 Baseline methods

Three baseline model architectures are tested, both on a set of synthetically generated catalogue structures and on the main PROCAT dataset.

Each method consists of a set encoding module and an attention-based pointing mechanism [Vinyals et al., 2015, Yin et al., 2020] for outputting the predicted permutation. The encoding module first obtains an embedding of individual offers through a recurrent neural network consisting of gated recurrent units [Chung et al., 2014] and then uses one of the three included methods of deriving the embedded representation of the entire set, which is permutation-invariant in 3 of the 4 cases.

The single exception to permutation invariance is a pure Pointer Network (1), which encodes the set sequentially through a stack of bidirectional LSTMs [Hochreiter and Schmidhuber, 1997, Schuster and Paliwal, 1997]. The remaining 3 methods are the Read-Process-Write model (2) [Vinyals et al., 2016], the Deep Sets encoder (3) [Zaheer et al., 2017] and the Set Transformer (4) [Lee et al., 2019].

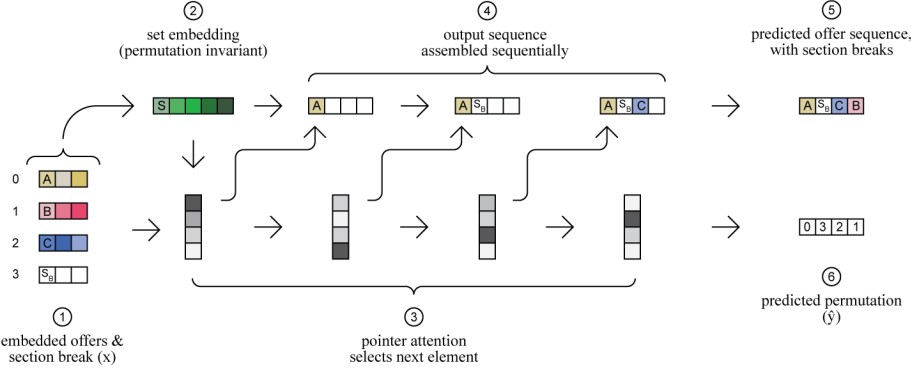

Figure 4: The input and output of the tested models, after the offer text embedding step.

In effect the random, shuffled order in which the available set of offers is originally presented to the model does not influence the representation of the set in methods 2, 3 and 4. The output of the attention-based pointing module is conditioned on this set representation through concatenating it with the embedding of each individual offer constituting the set. All models are implemented in

Table 2: Rank correlation coefficients for PROCAT

| Model | PROCAT | | Synthetic ($n = 20$) | |
|---|---|---|---|---|
| | Spearman $\rho$ | Kendall $\tau$ | Spearman $\rho$ | Kendall $\tau$ |
| Random Baseline | 0.004 | -0.01 | 0.09 | -0.07 |
| Pointer Network (2015) | 0.26 | 0.13 | 0.49 | 0.37 |
| Read-Process-Write (2016) | 0.30 | 0.18 | 0.52 | 0.41 |
| DeepSets (2017) | 0.35 | 0.22 | 0.55 | 0.44 |
| Set Transformer (2019) | **0.44** | **0.30** | **0.61** | **0.49** |

PyTorch following code written by their respective authors (where provided), and made publicly available on GitHub.

For a visual explanation of the input and output of the permutation-learning modules of the neural networks, see figure 4. The input to the compared models is always a list of raw-text documents representing offer instances, in a randomly permuted order that needs to be reverted to the target one.

## 4.2 Experimental setup and results

We perform experiments on an 80-20 training-validation split of the PROCAT dataset. Every model's weights are adjusted based on a cross entropy loss applied to the pointer attention vector over all set input elements at each step of the output sequence [Yin et al., 2020]. We use two rank correlation coefficients as our metrics, namely Spearman's rho ($s_\rho$):

$$s_\rho(y, \hat{y}) = 1 - \frac{6 \sum_{i=1}^{n} y_i - \hat{y}_i}{n(n^2 - 1)} \tag{1}$$

where $y$ is the target permutation in the form of integer ranks per element and $\hat{y}$ is the prediction; and Kendall's tau ($k_\tau$), which is calculated based on the number of concordant pairs between the target and predicted rank assignments [Shieh, 1998]. Additionally, we provide an aggregated percentage based correctness metric tracking how many elements per example input set were placed correctly.

Training on PROCAT is performed for 300 epochs with batch size of 64 using the Adam stochastic optimizer [Kingma and Ba, 2015] with a learning rate $10^{-4}$ and momentum 0.9. Each catalogue consists of $n = 200$ offers. Training on the synthetic dataset of 50,000 catalogue sequences of $n = 20$ elements is performed for 400 epochs with the same batch size and optimization hyperparameters, training on the synthetic dataset with sequences of $n \in \{15, 10\}$ is performed for 600 epochs, in an effort to show the feasibility of achieving better performance through the proposed, scaled-up set-to-sequence model architectures.

Every PROCAT model had a total of approximately 1 million trainable parameters, every model tested on the synthetic dataset had approximately 900 thousand. For details on the dimensions of layers, see the provided repository with code for repeated experiments.

An important implementation nuance comes in the form of progressive masking preventing the models from repeatedly pointing to the same element, which forces the output to be a valid permutation. It is also important to note that we do not currently directly measure the quality of clusters (sections) in PROCAT, and that whilst the target number of clusters varies per catalogue instance, that number is known to the model through the total count of section break tokens in the input set.

### 4.2.1 PROCAT results

Tables 2 and 3 present results for each of the 4 tested models and a baseline which always outputs valid but random permutations of the original input set. The final values of the Spearman's $\rho$ and Kendall's $\tau$ rank correlation coefficients are given for both the PROCAT dataset, with average cardinality of the input set (and therefore the length of the predicted permutation sequence) $n = 200$, and a sample of synthetic catalogue structures with $n \in \{20, 15, 10\}$. Metrics are averaged over 5 full training runs.

Overall, the models that obtain a permutation invariant representation of the set consistently perform better on the PROCAT dataset than a pure Pointer Network, which encodes the set sequentially through stacked RNNs. Furthermore, the top performing method has a built in mechanism for

Table 3: Rank correlation coefficients for synthetic datasets

| Model | Synthetic ($n = 15$) | | Synthetic ($n = 10$) | |
|---|---|---|---|---|
| | Spearman $\rho$ | Kendall $\tau$ | Spearman $\rho$ | Kendall $\tau$ |
| Random Baseline | -0.026 | -0.019 | 0.051 | 0.023 |
| Pointer Network (2015) | 0.67 | 0.54 | 0.73 | 0.61 |
| Read-Process-Write (2016) | 0.77 | 0.60 | 0.83 | 0.71 |
| DeepSets (2017) | 0.84 | 0.72 | 0.92 | 0.80 |
| Set Transformer (2019) | **0.96** | **0.85** | **0.98** | **0.93** |

encoding pairwise and higher-order interactions between set elements through transformer-style attention. Domain expertise suggests that interplay between individual product offers is indeed crucial when designing a product catalogue [Xu et al., 2013].

In figure 5 an analogous comparison of the average percentage of correctly predicted ranks per input set is given. Overall, the initial results are relatively low (under 7% for the Set Transformer), which illustrates the difficulty of the underlying task. Specifically, being able to predict a good section consisting of complementary offers but placing this section later in the output catalogue than in the original one would here be reflected with a 0% score regarding those elements. However, performance of the attention-based set encoder is more consistent, as indicated by narrower error bars.

Development of a more sensitive evaluation metric is both a direction for future work and the motivation behind the creation of the synthetic datasets, allowing for full control of the task's difficulty and more detailed insights into model performance.

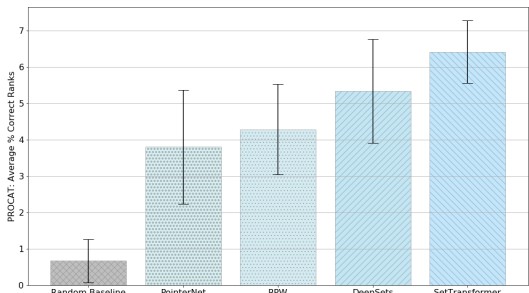

Figure 5: Comparison of the average percentage of correctly predicted ranks per input set element in the PROCAT dataset for the 4 main models and a random baseline, with error bars over 5 runs.

The fact that models which can explicitly encode higher-order interactions perform better suggests a range of future approaches. These could include: using the provided priority class information that encodes visual offer placement information; applying learn-to-rank frameworks with the set representation as the query for which offer relevance is determined; and exploring the possibility of predicting catalogues as directed graphs, particularly ones consisting of disjoint cliques guaranteeing a valid clustering [Serviansky et al., 2020].

### 4.2.2 Functional results on synthetic data

The results for synthetic datasets consisting of 50,000 simplified catalogue structures of lengths $n_i \in \{20, 15, 10\}$, generated following the challenging default set of clustering and structural rules, are given in the right half of table 2 as well as in tables 3 and 4. All results are averaged over 5 full training and testing runs.

The results for functional tests for reporting model performance per rule and type of rule in table 4 are of particular interest. These have been aggregated into the *clustering* score, which is the average percentage of valid sections per catalogue (based on default section rules), the *structural* score, which is the average percentage of predicted catalogues following the structural (section order) rules that do not depend on pairwise or higher order interactions between input set elements, and finally *structural*$^{2+}$, which relates to structural rules that do.

Table 4: Functional tests

| Model | Synthetic ($n = 20$) | | | Synthetic ($n = 15$) | | |
|---|---|---|---|---|---|---|
| | Clustering | Structural | Structural $^{2+}$ | Clustering | Structural | Structural $^{2+}$ |
| Random Baseline | 0.08 | 0.03 | 0.01 | 0.09 | 0.03 | 0.02 |
| Pointer Network (2015) | 0.39 | 0.21 | 0.13 | 0.61 | 0.53 | 0.29 |
| Read-Process-Write (2016) | 0.40 | 0.25 | 0.13 | 0.64 | 0.45 | 0.34 |
| DeepSets (2017) | 0.43 | 0.35 | 0.16 | 0.75 | 0.61 | 0.37 |
| Set Transformer (2019) | **0.63** | **0.57** | **0.32** | **0.89** | **0.88** | **0.75** |

Overall, in terms of the clustering score, i.e. whether the section composition in predicted catalogues followed the rules from the synthetically generated ones, the difference in performance between methods that obtain a permutation invariant representation of the input set and those that do not was less pronounced than in terms of the two structural scores. It is unclear as to why this occurs, as both section composition and section order are defined by the composition of the input set.

Nonetheless, the model capable of explicitly encoding pairwise and higher order interactions between input set elements (4) outperforms the rest in terms of the *structural*$^{2+}$ score, predicting catalogues abiding by such structural rules in 32% of cases for $n = 20$ and 75% of cases for $n = 15$, showcasing a significant impact of set cardinality and sequence length on model performance.

## 4.3 Computational resources

The experiments were performed on cloud-based GPU instances provisioned from the Paperspace computing platform, with NVIDIA Quadro P6000 graphics cards (24 GB) and 8 CPU cores. Following the carbon emission calculator developed by Lacoste et al. [2019], we estimate the total $CO^2$ emissions for all performed experiments at 32.4 kg, and the cost of training the best performing model at 1.08 kg (over 10 hours).

Whilst the Paperspace cloud platform does not provide specific information about how much of its infrastructure's energy consumption it offsets, it is worth noting that one of the goals of solving the set-to-sequence catalog prediction task is to reduce paper waste by making physical catalogues obsolete. Thus it is hard to calculate the final impact on $CO^2$ emissions [Pivnenko et al., 2015].

## 4.4 Ethical considerations and societal impact

Given the e-commerce context of the main presented dataset, we must highlight the wider problem of *endless scroll* user interfaces in product presentation apps and social media [Lupinacci Amaral, 2020].

Whilst the PROCAT dataset is only tailored to predicting finite-length sequences from sets, we cannot rule out the possibility of extending set-to-sequence models to non-finite sets. It is also in principle possible to retrain the discussed models with additional inputs in the form of e.g. embedded personal preferences, making the predicted catalogs tailored to specific individuals, which has been linked to mental health issues related to smartphone addiction [Noë et al., 2019].

In an effort to mitigate this risk, we did not include any user interaction information; doing so could indicate the performance of individual catalogues in terms of user engagement. This information was excluded despite it being likely to signal optimal catalogue structures, as indicated by case studies in the field of ML classification [Ferrari et al., 2020] and clinical decision support [Chen et al., 2020]. As a consequence, the dataset contains no personal information and is GDPR-compliant.

We do not see any clear way for it to exacerbate bias against people of a certain gender, race, sexuality, or who have other protected characteristics. However, it may be without merit to consider bias that may have been inherent to the marketing decisions made by people who have designed the catalogues contained in the dataset, such as the *pink tax* [Stevens and Shanahan, 2017].

## 5 Limitations

The PROCAT dataset consists of text in Danish, which has only six million users. However, this can also be seen as a benefit in terms of providing domain-specific, publicly available resources for a

non-privileged language [Kirkedal et al., 2019]. The catalogue ordering problem is independent of language, so we consider this limitation to be of low impact.

An important limitation of PROCAT and learning from human-made product catalogues in general, is that we only have access to one canonical ordering of the offer instances, whereas it is not impossible that other, equally valid catalogues can be constructed from the same input set of offers. In order to mitigate this, we provide the synthetic dataset library, where many valid permutations are available for each input set, increasing the signal to noise ratio.

The benchmark methods provided with PROCAT take a single-step approach. It is not currently clear whether a single step approach to predicting the product catalogue structure in a set-to-sequence formulation is the most viable. Other, multi-stage approaches might circumvent the problem of handling the padding used in the presented version of PROCAT, increasing the signal-to-noise ratio in the dataset. It is possible to use the currently provided raw data for other formulations of the underlying task.

## 6  Conclusion

We have highlighted the need for and provided a publicly available, easily accessible and reliably maintained product catalogue dataset. The value of the dataset stems from the difficulty of the structure prediction task, which involves representation learning, implicit clustering and permutation learning challenge. This motivates experiments with models capable of predicting complex structures as presented in sections 2.1 and 4.1.

We address the need for such a data source by curating PROCAT – a dataset of over 10,000 expert-designed product catalogues consisting of more than 1.5 million individual product offers, grouped into complementary sections. Additionally, due to the complexity of the underlying data, we also provide a library for generating simplified synthetic catalogues according to chosen clustering and structural rules. The performance of the proposed models is then measured per rule, allowing for a more fine-grained look into what our models have actually learned, through functional tests.

Benchmarks indicate that the PROCAT structure prediction task is considerably difficult. Attention-based models capable of explicitly encoding pairwise and higher order interactions between set elements outperform other set encoders and pure permutation learning models. We believe there are other interesting tasks and methods PROCAT may inspire, though an in-depth exploration is beyond the scope of this dataset paper.

We intend to improve and expand both the PROCAT dataset and the synthetic data generation library in order to facilitate the development of practical solutions in intelligent, privacy-centric product presentation systems.

## Acknowledgements

This work was partly supported by an Innovation Fund Denmark research grant (number 9065-00017B) and by Tjek A/S. The authors would like to acknowledge Rasmus Pagh's assistance in model design and benchmark task conceptualization.

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
