# OpenReview forum: "PROCAT: Product Catalogue Dataset for Implicit Clustering, Permutation Learning and Structure Prediction"
_NeurIPS.cc/2021/Track/Datasets_and_Benchmarks/Round1 — NeurIPS 2021 Datasets and Benchmarks Track (Round 1)_

### Official Review · Reviewer_qGaZ · 2021-07-04
**PROCAT: Product Catalogue Dataset for Implicit Clustering, Permutation Learning and Structure Prediction**

**Rating:** 5
**Confidence:** 4
**Clarity:** The paper is well written.

**Strengths:**

The introduction of data collection, statistical information, and application in this paper is very detailed. And the paper is well written.

**Weaknesses:**

I have two concerns.
First, the diversity of the dataset is relatively limited. Because the language and number of users involved in the data collection process are not enough to show the diversity of the dataset.
Then, whether the challenge of the dataset that the authors emphasize is caused by the quality of the dataset.

**Additional Feedback:**

See the discussion in "Weaknesses".

**Correctness:**

The paper did not propose a benchmark method, and the baseline methods for verification are also very limited.

**Documentation:**

Yes

**Ethics:**

See the discussion in "Weaknesses".

**Relation To Prior Work:**

The comparison with related work is very limited in this paper. This is not enough to highlight the difference and contribution of the paper relative to the previous work.

**Summary And Contributions:**

The paper proposes an e-commerce dataset consisting of over 1.5 million set items across a 4-year period, in both raw text form and with pre-processed features containing information about relative visual placement.

---

> ### Author Response · Authors · 2021-07-12
> **Response to Reviewer qGaZ (part 2)**
>
> **Q3: "The paper did not propose a benchmark method, and the baseline methods for verification are also very limited."**
>
> We are uncertain as to how to interpret this comment and would appreciate clarification.
>
> We have provided a benchmark task (set-to-sequence catalogue structure prediction) for both the main dataset and the synthetic ones, including detailed per-rule metrics. All code for repeated experiments and the best performing model is available at:
>
> https://github.com/mateuszjurewicz/procat
>
> We have tested the following three state-of-the-art [3] set encoder models combined with a permutation learning module on the above-mentioned datasets:
>   * Pointer Network [4][5]
>   * DeepSets [6][7]
>   * Set Transformer [8]
>
> In an effort to expand upon the originally provided benchmark methods we have also included the performance of the previously omitted Read-Process-and-Write architecture [9], the performance of a randomized baseline and the performance of all tested models on synthetic datasets consisting of catalogue structures of different cardinalities.
>
> **Q4: "The comparison with related work is very limited in this paper. This is not enough to highlight the difference and contribution of the paper relative to the previous work."**
>
> We thank the reviewer for pointing this aspect of the paper to us. We have adjusted the text to better reflect that the proposed dataset is (to the best of our knowledge) the only existing product catalogue structure dataset providing offer placement and section order information and expanded the section devoted to related methods.
>
> * [3] Skianis, K., Nikolentzos, G., Limnios, S., & Vazirgiannis, M. (2020, June). Rep the set: Neural networks for learning set representations. In International conference on artificial intelligence and statistics (pp. 1410-1420). PMLR.
> * [4] Vinyals, O., Fortunato, M., & Jaitly, N. (2015). Pointer networks. Advances in Neural Information Processing Systems, 28, 2692–2700.
> * [5] Vinyals, O., Babuschkin, I., Czarnecki, W. M., Mathieu, M., Dudzik, A., Chung, J., ... & Silver, D. (2019). Grandmaster level in StarCraft II using multi-agent reinforcement learning. Nature, 575(7782), 350-354.
> * [6] Zaheer, M., Kottur, S., Ravanbhakhsh, S., Póczos, B., Salakhutdinov, R., & Smola, A. J. (2017). Deep sets. Advances in Neural Information Processing Systems, 30, 3392–3402.
> * [7] Hemati, S., Kalra, S., Meaney, C., Babaie, M., Ghodsi, A., & Tizhoosh, H. (2021, February). CNN and Deep Sets for End-to-End Whole Slide Image Representation Learning. In Medical Imaging with Deep Learning.
> * [8] Lee, J., Lee, Y., Kim, J., Kosiorek, A., Choi, S., & Teh, Y. W. (2019, May). Set transformer: A framework for attention-based permutation-invariant neural networks. In International Conference on Machine Learning (pp. 3744-3753). PMLR.
> * [9] Vinyals, O., Bengio, S., & Kudlur, M. (2016). Order matters: Sequence to sequence for sets. 4th International Conference on Learning Representations, ICLR 2016 - Conference Track Proceedings, 1–11.

---

> ### Author Response · Authors · 2021-07-12
> **Response to Reviewer qGaZ (part 1)**
>
> Thank you for your review and the provided comments. Please find our responses to all points raised below, along with specific adjustments to the original paper intended to address them (available in the revised version).
>
> **Q1: "I have two concerns. First, the diversity of the dataset is relatively limited. Because the language and number of users involved in the data collection process are not enough to show the diversity of the dataset."**
>
> *This is a concern mentioned by one other reviewer, therefore the two responses are similar. We are providing them in both places with small adjustments for ease of reading & responding.*
>
> We agree that the diversity of the dataset is limited due to the offer text being in Danish. Our intention was to provide a valuable resource for an underrepresented language. One aspect of the dataset that we failed to mention in the paper is that the catalogues come from a wide variety of providers, including cross-border shops that have a significant following in neighboring Scandinavian countries, particularly Sweden and Norway, as well as Germany.
>
> We have also adjusted the paper to reflect the diversity of the dataset by providing an overview of commercial categories that the catalogues belong to, following the Global Product Classification (GPC-GS1) with multiple categories per catalogue, included below for simplicity. We hope that our paper can provide a simple, easy-to-use format for releasing product catalogue datasets in other languages.
>
> | Category   	   	|      Number of Catalogues     |  % 	|
> |-----------------------|-------------------------------|-------|
> | Food (FBT)		| 7456				| 67.40%|
> | Electronic		| 5231				| 47.28%|
> | Personal Care		| 5113				| 46.22%|
> | Tools			| 3311				| 29.93%|
> | Sports Equipment	| 2147				| 19.41%|
> | Lawn/Garden Supplies	| 2039				| 18.43%|
> | Home Appliances	| 2028				| 18.33%|
> | Baby Care		| 1986				| 17.95%|
> | Household Furniture	| 1672				| 15.11%|
> | Pet Care		| 1522				| 13.76%|
> | Footwear		| 1324				| 11.97%|
> | Toys and Games	| 1293				| 11.69%|
> | Fuels			| 548				| 4.95% |
>
> We are not certain what is meant by the number of users involved in the data collection process and would appreciate clarification. Nonetheless, we have enhanced the paper by providing information about the number of individual retailers that the catalogues belonged to (2,400) and the total number of unique users who have viewed the catalogues within the app (2.5 million).
>
> **Q2: "(concern) whether the challenge of the dataset that the authors emphasize is caused by the quality of the dataset."**
>
> We agree that the core dataset presents a significant challenge. We believe this charcteristic to be inherent to product catalogue structure prediction, reflecting the complexity of how this process is currently performed by human experts.
>
> We have revised the paper to better highlight the fact that through the provided synthetic datasets we allow other researchers to fully control the difficulty of the proposed set-to-sequence tasks.
>
> Our hope is that initial testing of models can be performed on the synthetic datasets, making it possible to distinguish between the difficulty related to permutation learning and the difficulty inherent to the real-world dataset itself.

---

### Official Review · Reviewer_ado7 · 2021-07-05

**Rating:** 7
**Confidence:** 4

**Strengths:**

- The new dataset is large scale, with over 1.5 millions items collected over the course of 4 years, and collected from 10k different catalogues (offering diversity). The items and offers are augmented by multiple fields, such as prices, sections, priority classes, and ordering.
- The dataset has multiple potential practical applications and uses in the field of e-commerce and is thus a great contribution to the research community. The paper also does great in terms of motivating the need for the dataset and describing these potential uses, both from commerce perspective, as well as in terms of challenges for the machine learning community.
- It's great to see that the authors show awareness and work to address ethical aspects related to constructing and releasing the dataset (more details below).
- As discussed above it's good that the dataset comes with performance analysis of baseline models for related prediction tasks, and in addition to that, it also provides a library for auto-generation of synthetic data that supports new evaluation metrics. I find the balance between having both real-world data but also providing means for a more controlled analysis and evaluation through synthesized data to be both practically useful and academically interesting.
- The paper does well in presenting in detail both the collection and preparing process of the data, and what each items


**Weaknesses:**

- I find it a bit surprising that for the experiments the authors choose to focus only on the set-to-sequence prediction use of the dataset. This sounds like a very specific task, while I'm certain there could be a larger variety of supervised and unsupervised tasks on top of the data, in the areas of clustering, generating textual descriptions given the product image, estimating prices, predicting recommendation for new items based on previously purchased ones, etc etc.
- Following up on the previous point, for some of these tasks, additional annotation within the dataset will be necessary. I think that the underlying dataset is a great and quite large resource, and it could be really nice if it was augmented with additional annotations of the items (some potential examples: prices, recommendations, properties of users that bought them, correlation between purchasing different items, etc. etc.). I don't know if these details could be easily collected but would be great to explore this direction!
- Not detailed enough online documentation of the dataset. Will be helpful to have a website with some information about the dataset, its key features, and the files/directory structure and format.

**Additional Feedback:**

Nope, all my feedback is above.

**Clarity:**

The paper is clear, easy-to-follow and written in an interesting compelling manner. The writing quality is good and there are multiple helpful visualizations. It would be helpful if the website did a better job in presenting the dataset and providing more details about its format and main features.

**Correctness:**

The claims in the paper seem to me sound and they are corroborated by details, explanations, experiments (for the baseline models), and examples.

**Documentation:**

A more accessible easy-to-use website that describes the dataset and explains the structure and format of the files in it etc could be very useful.

**Ethics:**

There are potential ethical and social aspects in terms of e.g. privacy in using personal information, but the paper explicitly discuss them and explore ways to mitigate them, and so does a good job in addressing these concerns. It furthers clearly provides information about sustainability and licensing and also discusses limitations of the work.

**Relation To Prior Work:**

The paper does a good job in comparing to prior work and contextualizing the new dataset within the current field. It motivates the general task, provides background in the area including prior works, and explaining how the new dataset is distinguished from them.  It also discusses various set-to-sequence methods (which are relevant for the experiments section).

**Summary And Contributions:**

The paper presents a new large-scale dataset for products catalogue with both textual descriptions and structured information as well as meaningful ordering between the items in the catalogue. I'm not aware of many public datasets with structured/ordered information in this realm and so that is great to have a new one that covers that. In addition, it adds also baseline experiments on the task of set-to-sequence prediction over the data as well as means to generate additional synthetic catalogue data. Finally, The overall paper is written clearly and the work is well-presented, motivated and justified.

Update: After reading the author's response I would like to keep my score and overall I think it reflects my thoughts well!

---

> ### Author Response · Authors · 2021-07-12
> **Response to Reviewer ado7**
>
> Thank you for your thoughtful review and encouraging comments.
>
> **Q1: "I think that the underlying dataset is a great and quite large resource, and it could be really nice if it was augmented with additional annotations of the items."**
>
> We agree that enriching the data with extracted price, quantity per unit and other information of this type will be valuable. The price information is currently found within the raw text belonging to an offer but we are planning to include it in extracted form in the future, along with e.g. price per unit.
>
> **Q2: "It would be helpful if the website did a better job in presenting the dataset and providing more details about its format and main features. A more accessible easy-to-use website that describes the dataset and explains the structure and format of the files in it etc could be very useful."**
>
> We have enhanced both the figshare website and the github repository containing the code for repeated experiments with a more detailed explanation of the dataset, its format, features and file structure:
>
> https://github.com/mateuszjurewicz/procat#dataset-structure

---

> > ### Comment · Reviewer_ado7 · 2021-07-20
> > **Thanks!**
> >
> > Thanks for the response and for making the github repo!

---

### Official Review · Reviewer_kfTg · 2021-07-06
**Not a well-defined problem formulation**

**Rating:** 4
**Confidence:** 3
**Correctness:** The experiments and evaluation method…

**Strengths:**

1. This dataset is the first of its kind and can open up new research directions.
2. The dataset contains more than 1.5 million individual product offers which can be used for training large neural networks.
3. Basic baselines are provided for both real and synthetic datasets, and the code is also shared.


**Weaknesses:**

1. Only one "right" order is provided for each product catalog. At the same time, there are usually many reasonable ways of designing the catalogs, and the human-designed one might be far from optimal. This formulation makes the problem too hard. Even on the synthetic dataset, the performance is very low.
2. The dataset is not diverse as it only includes text in Danish. However, designing product catalogs is a very subjective task, and the preferences might be widely different in other parts of the world.
3. The image of the products are not included in this dataset. Sometimes the relative ordering of the product offers in the same section is determined by their pictures (e.g., similar-looking products might be located close to each other). Without that information finding the optimal ordering seems infeasible.

**Additional Feedback:**

The authors can make their dataset and work stronger by including:
1. More than one alignment for each product catalog (for example, three possible orderings by three different human designers)
2. Product catalogs in other languages and other parts of the world to ensure a diverse dataset.
3. Product photo for each offer to better predict the relative order within sections.

**Clarity:**

The paper is mostly clear. However, the authors could do a much better job in including better figures. The main structure of the dataset and the input outputs of the neural network are not clearly explained.

**Documentation:**

The appendix helps for better understanding the details and documenting the process.


**Ethics:**

One straightforward application is more targeted advertisements which probably only benefits large cooperates.

**Relation To Prior Work:**

This is a new task and a new dataset designed for it, but related datasets are discussed in section 2.2.

**Summary And Contributions:**

The authors present PROCAT dataset containing 11,063 product catalogs. Each product catalogs consist of several sections (pages) and many product offers. This dataset is made up of 1,613,686 product offers (instances). Each offer instance consists of its unique id, related section, catalog ids, text heading, and description. The proposed task is a set-to-sequence problem, where the ordered list of offers (the ranking) is predicted given the set of offer descriptions. The main contributions are:
1. Defining a new task for learning the correct permutation of offers in product catalogs.
2. Providing a new dataset containing more than 1 million product offers.
3. Introducing a synthetic dataset for the same task.

---

> ### Author Response · Authors · 2021-07-12
> **Response to Reviewer KFTG (part 2)**
>
> **Q3: "The images of the products are not included in this dataset. Sometimes the relative ordering of the product offers in the same section is determined by their pictures (e.g., similar-looking products might be located close to each other). Without that information finding the optimal ordering seems infeasible."**
>
> It is very valid to note that visual representation of offers could be helpful in solving the catalogue prediction tasks. However, multiple meetings with SMEs in a number of leading retail companies have reveled a consistent (if surprising) trend of the catalogue structure being determined before graphic designers create the final product images, precluding their impact. This, along with technical and copyright concerns motivated us to exclude them in this iteration of the dataset.
>
> Additionally, given the purpose of the paper being to facilitate the creation of automated product catalogues that can be rendered on any device, the relative positioning of offers within a section will differ depending on available screen size. Instead, the provided priority category per offer is what defines the in-section positioning.
>
> **Q4: "The paper is mostly clear. However, the authors could do a much better job in including better figures. The main structure of the dataset and the input outputs of the neural network are not clearly explained."**
>
> We'd like to thank the reviewer for pointing this out. We have improved several figures in the paper and added a separate figure outlining the flow of offer representations through the permutation learning module in a separate figure (number 4), to clearly state the input and output of the proposed neural networks.

---

> > ### Comment · Reviewer_kfTg · 2021-07-15
> > **Ambiguous and limited dataset**
> >
> > I would like to thank the authors for taking the time and responding to the reviewers. The new version of the paper is improved and provides a better illustration of their dataset and method. While I appreciate the author's points, I still believe that the dataset in its current form is too ambiguous and limited.  Designing catalogs is a very subjective task, and providing only a single right arrangement without sharing the pictures of the products seems too vague to get meaningful signals. It is surprising to know that catalog structure being determined before graphic designers create the final product images; however, it is only helpful when graphic design happens after designing catalog structure by experts. Suppose we want the AI model to develop the whole catalog; it also needs to take into account the pictures and arrange the images appropriately. Otherwise, still, a human expert is required at the end to do the graphic design.
> >
> > Global Product Classification is good additional information, and based on this information, I think the dataset might be diverse enough for being used in Scandinavian countries. However, I still don't think it would be helpful globally since consumer's language, culture, preferences, and even type of products are widely diverse in different parts of the world.
> >
> > Based on the ambiguity of the task and dataset and its limits in diversity, I don't change my rating, and I cannot recommend this work for publication at this point.

---

> ### Author Response · Authors · 2021-07-12
> **Response to Reviewer KFTG (part 1)**
>
> Thank you for a thoughtful and insightful review. Please find the responses to the points you have raised below, along with specific adjustments to the original paper intended to address them (available in the revised version).
>
> **Q1: "Only one 'right' order is provided for each product catalog. At the same time, there are usually many reasonable ways of designing the catalogs (...). This formulation makes the problem too hard. Even on the synthetic dataset, the performance is very low."**
>
> Yes, only one valid order per catalogue is provided. We agree that this aspect of the dataset presents a significant core challenge. In practice, we unfortunately never have access to multiple human-designed catalogues formed from the same underlying set of offers.
>
> It is also true that this formulation makes evaluation of the predicted catalogues challenging. Our idea was to adopt a framework similar to the one often used in NLP for evaluating language models [1][2]. It includes training models on real text input and then evaluating them via the similarity of the predicted text samples to the input data, as measured through perplexity. We believe our current task formulation to be analogous.
>
> We have adjusted the paper to better highlight the fact that *through the provided synthetic datasets we do give the models access to multiple valid catalogue structures that are obtained from the same input set*. The synthetic generation library gives researchers full control over the difficulty of the task, as explained in the repository for repeated experiments:
>
> https://github.com/mateuszjurewicz/procat
>
> Thank you for letting us know about this shortcoming. To further mitigate it, we have also provided results for synthetic sets of cardinalities other than n=20, highlighting the better performance of proposed models in these settings after retraining for more epochs. We would additionally like to point to the best proposed model's performance in terms of the rank correlation coefficient (Spearman's Rho) where on a scale of -1 to 1 our initial model reached 0.44, which we believe to be a promising early result. We believe this to be a better evaluation metric than the binary positional accuracy.
>
> * [1] Guo, M., Dai, Z., Vrandečić, D., & Al-Rfou, R. (2020, May). Wiki-40b: Multilingual language model dataset. In Proceedings of The 12th Language Resources and Evaluation Conference (pp. 2440-2452).
> * [2] Meister, C., & Cotterell, R. (2021). Language Model Evaluation Beyond Perplexity. arXiv preprint arXiv:2106.00085.
>
> **Q2: "The dataset is not diverse as it only includes text in Danish. However, designing product catalogs is a very subjective task, and the preferences might be widely different in other parts of the world."**
>
> We agree that the diversity of the dataset is limited due to the offer text being in Danish. Our intention was to provide a valuable resource for an underrepresented language. One aspect of the dataset that we failed to mention in the paper is that the catalogues come from a wide variety of providers, including cross-border shops that have a significant following in neighboring Scandinavian countries, particularly Sweden and Norway, as well as Germany.
>
> We have also adjusted the paper to reflect the diversity of the dataset by providing an overview of commercial categories that the catalogues belong to, following the Global Product Classification (GPC-GS1) with multiple categories per catalogue, included below for simplicity. We hope that our paper can provide a simple, easy-to-use format for releasing product catalogue datasets in other languages.
>
> | Category   	   	|      Number of Catalogues     |  % 	|
> |-----------------------|-------------------------------|-------|
> | Food (FBT)		| 7456				| 67.40%|
> | Electronic		| 5231				| 47.28%|
> | Personal Care		| 5113				| 46.22%|
> | Tools			| 3311				| 29.93%|
> | Sports Equipment	| 2147				| 19.41%|
> | Lawn/Garden Supplies	| 2039				| 18.43%|
> | Home Appliances	| 2028				| 18.33%|
> | Baby Care		| 1986				| 17.95%|
> | Household Furniture	| 1672				| 15.11%|
> | Pet Care		| 1522				| 13.76%|
> | Footwear		| 1324				| 11.97%|
> | Toys and Games	| 1293				| 11.69%|
> | Fuels			| 548				| 4.95% |
>
> Finally, we have provided additional information about the number of individual retailers that the catalogues belonged to (2,400) and the total number of unique users who have viewed the catalogues within the app (2.5 million). Our hope was to represent a broad array of product categories and providers.

---

### Note · ~Mateusz_Maria_Jurewicz1 · 2021-07-12

https://github.com/mateuszjurewicz/procat

---

### Decision · Program_Chairs · 2021-07-27

**Decision:**

Accept

**Comment:**

In this paper the authors offer a dataset of many product catalogues and the task of predicting the order of products based on the set of items and their features. The task is fairly novel and the dataset is notably quite large. As with many datasets containing text, it could be improved by covering a broader swath of languages. Further, there were ideas raised of how to improve or expand the uses of this dataset, e.g. for new tasks or to allow flexibility beyond a single correct answer, and these would likely be valuable improvements.

In discussing with other ACs, we agreed that while the paper could be improved by covering more languages, this dataset is not unique in only covering one language and thus not grounds for rejection. With respect to including more than a single "correct" ranking per example, we felt that the scale of the dataset outweighed this downside. It'd be great if the authors could improve along these dimensions, but even with these limitations we feel the benefits outweigh the limitations.